# Emerging Mutations in Nsp1 of SARS-CoV-2 and Their Effect on the Structural Stability

**DOI:** 10.3390/pathogens10101285

**Published:** 2021-10-06

**Authors:** Kejie Mou, Farwa Mukhtar, Muhammad Tahir Khan, Doaa B. Darwish, Shaoliang Peng, Shabbir Muhammad, Abdullah G. Al-Sehemi, Dong-Qing Wei

**Affiliations:** 1Department of Neurosurgery, Bishan Hospital of Chongqing, Chongqing 402760, China; kjb@bsrmyy.cn; 2Institute of Molecular Biology and Biotechnology (IMBB), The University of Lahore, KM Defence Road, Lahore 58810, Pakistan; pbt02203006@student.uol.edu.pk; 3Botany Department, Faculty of Science, Mansoura University, Mansoura 35516, Egypt; ddarwish@ut.edu.sa; 4Department of Biology, Faculty of Science, University of Tabuk, Tabuk 71491, Saudi Arabia; 5Peng Cheng Laboratory, Vanke Cloud City Phase I Building 8, Xili Street, Nashan District, Shenzhen 518055, China; slpeng@hnu.edu.cn; 6Department of Physics, College of Science, King Khalid University, Abha 61413, Saudi Arabia; mshabbir@kku.edu.sa; 7Research Center for Advanced Materials Science (RCAMS), King Khalid University, Abha 61413, Saudi Arabia; agsehemi@kku.edu.sa; 8Department of Chemistry, College of Science, King Khalid University, Abha 61413, Saudi Arabia; 9State Key Laboratory of Microbial Metabolism, Shanghai-Islamabad-Belgrade Joint Innovation Center on Antibacterial Resistances, Joint International Research Laboratory of Metabolic & Developmental Sciences and School of Life Sciences and Biotechnology, Shanghai Jiao Tong University, Shanghai 200030, China

**Keywords:** SARS-CoV-2, genome, mutations, Nsp1

## Abstract

The genome of severe acute respiratory syndrome coronavirus 2 (SARS-CoV-2) encodes 16 non-structural (Nsp) and 4 structural proteins. Among the Nsps, Nsp1 inhibits host gene expression and also evades the immune system. This protein has been proposed as a target for vaccine development and also for drug design. Owing to its important role, the current study aimed to identify mutations in Nsp1 and their effect on protein stability and flexibility. This is the first comprehensive study in which 295,000 complete genomes have been screened for mutations after alignment with the Wuhan-Hu-1 reference genome (Accession NC_045512), using the CoVsurver app. The sequences harbored 933 mutations in the entire coding region of Nsp1. The most frequently occurring mutation in the 180-amino-acid Nsp1 protein was R24C (*n* = 1122), followed by D75E (*n* = 890), D48G (*n* = 881), H110Y (*n* = 860), and D144A (*n* = 648). Among the 933 non-synonymous mutations, 529 exhibited a destabilizing effect. Similarly, a gain in flexibility was observed in 542 mutations. The majority of the most frequent mutations were detected in the loop regions. These findings imply that Nsp1 mutations might be useful to exploit SARS-CoV-2′s pathogenicity. Genomic sequencing of SARS-CoV-2 on a regular basis will further assist in analyzing variations among the drug targets and to test the diagnostic accuracy. This wide range of mutations and their effect on Nsp1’s stability may have some consequences for the host’s innate immune response to SARS-CoV-2 infection and also for the vaccines’ efficacy. Based on this mutational information, geographically strain-specific drugs, vaccines, and antibody combinations could be a useful strategy against SARS-CoV-2 infection.

## 1. Background

Severe acute respiratory syndrome 2 (SARS-CoV-2) is responsible for one of the most serious pandemics in human history and has caused huge human and financial losses. There is an urgent need for a molecular framework to determine how the virus deploys the host cellular machinery. Continuous molecular characterization of virus genomes from different geographic regions should be investigated to analyze the factors involved, particularly the virus’ pathogenicity and its evolutionary phases. 

The genome of SARS-CoV-2 encodes 16 non-structural proteins (Nsps) and 4 structural proteins [1]. ORF1ab, the largest component of the virus genome, harboring a high frequency of mutations, splits into 16 Nsps [2,3]. Among these, Nsp1 anchors the replication complex to the cellular membranes as a membrane-associated host translation inhibitor. It also inhibits host translation by forming a complex with a 40S ribosomal subunit. Nsp1 degrades the host mRNAs by prompting endonucleolytic cleavage near the 5’-untranslated region [4]. Due to the presence of the 5´-end leader sequence, SARS-CoV-2 mRNAs are not prone to Nsp1-mediated endonucleolytic degradation. The leader protein advances the proficient articulation of viral genes by suppressing the host gene expression in infected cells and inhibiting the host immune response [5].

The Nsp1 of SARS-CoV-2 contains a globular domain (amino acids 13 to 121) between disordered residue locations from 1 to 12 and 122 to 179 [6]. The crystal structure of Nsp1 (PDB ID: 7k7p) consists of amino acid (aa) residues from 10 to 126. The regular Nsp1 secondary structure consists of β1 (aa13-20), followed by β2 (aa51-54), β3 (aa68-73), β4 (aa84-92), β5 (aa95-97), -β6 (aa103-109), and β7 (aa117-123), with 3_10_ helixes (η1 and η2). η1 (aa23-25) and η2 (aa61-63) are located at the opening, across one barrel, and α1 (aa34-49) is positioned alongside the barrel in contact with β4 [6]. In a recent study, the structure of SARS-CoV-2 Nsp1 (aa13-127) was shown to have a feature arrangement of a six-stranded (*n* = 6) β-barrel, antiparallel except for strands β1 (Q15–V20) and β2 (C51–V54) [7]. A unique feature of SARS-CoV-2 Nsp1 is the presence of a large number of flexible loops.

Nsp1 is the N-terminal product of ORF1ab, cleaved by papain-like proteinase (PLpro) from the polyprotein, promoting host mRNA degradation, blocking translation, and interfering with the innate immune response [8,9,10,11]. It also binds with ribosomal subunits (40S), inserting its C-terminal region into the entrance position of the channel (ribosomal mRNA), blocking the translation of mRNA [12]. The shutdown of the translational process inhibits the innate immune response [13]. Potential drugs may be designed to target the binding pocket, present on the ribosome. The deletion of the Nsp1-coding region results in preventing the cultured cells from becoming infected [6]. Furthermore, mutations in the coding region of Nsp1 may prevent its release from the nascent polyprotein ORF1a, limiting the viability of the virus. More recently [14], mutations in Nsp1 and their association with clinical phenotypes showed that an in-frame deletion in the N-terminal domain is associated with lower IFN-β levels in infected patients. The 500-532 region is the most frequent deletion locus across the genome of SARS-CoV-2. The biological importance of the deletion lies in the reduced type I interferon response, highlighting that a mutation in Nsp1 may also affect the host immune response.

We investigated the mutations’ frequency and their possible effects on Nsp1 from experimentally determined whole-genome sequences, submitted to the Global Initiative Sharing All Influenza Data (GISAID). These globally collected genome sequences were compared with the SARS-CoV-2 reference genome. This is the first comprehensive study in which 295,000 complete genomes of SARS-CoV-2 were screened for mutations in all amino acid positions of Nsp1. All detected mutations were subjected to analysis of their stability and flexibility effects on Nsp1’s structure using the DynaMut server.

## 2. Methodology

### 2.1. Genome Sequence Retrieval

The SARS-CoV-2 genome sequences were retrieved from GISAID (Dec, 2019 to Dec, 2020) (https://www.gisaid.org/, accessed on 30 December 2020) [15]. GISAID shares SARS-CoV-2 genomic data to facilitate their analysis for the purpose of public health and it published both full results and metadata to meet the requirements of public health scientists. A total of 295,000 whole-genome sequences (WGS) have been reported worldwide since December 2020. These WGS data can be downloaded in FASTA format for further analysis, including mutation screening, phylogenetics, and also emerging evolutionary phases of SARS-CoV-2. The GISAID server also provides some basic information regarding sequences for data scientists. 

### 2.2. Genome Sequence Alignment 

All the WGS sequences were aligned with the reference genome of Wuhan-Hu-1 (Accession NC_045512) using the CoVsurver application (https://www.gisaid.org/epiflu-applications/covsurver-mutations-app/, accessed on 30 December 2020). The CoVsurver is an app of GISAID that aligns query sequences with a reference genome, and the results are displayed in the form of amino acid mutations at specific locations in all structural and Nsp proteins. 

All the detected mutations in the Nsp1 of SARS-CoV-2 were organized in Excel sheets. Statistical analysis was performed using EpiData Analysis V2.2.3.187 [16] to screen the most common mutations. EpiData is widely applied to analyze large datasets. The World Health Organization (WHO) uses EpiData for the biostatistical analysis of epidemiological and public health data, and also for other quantitative analyses.

### 2.3. Nsp1 Structural Information

The sharing of infectious agent genomic data is critical to fight against disease outbreaks. The SARS-CoV-2 structural data were retrieved from the Protein Data Bank (PDB) (PDB ID: 7k7p) [17]. PDB (http://www.rcsb.org/pdb/, accessed on 30 December 2020) is a single structural data archive of biological molecules. All the data are primary, collected from depositors across the globe. The data on the macromolecules also contain the coordinates, structures, and methods of structural determination. 

### 2.4. Missing Residues and Full-Length Nsp1

The PDB structure of Nsp1 has some missing residues; therefore, we retrieved the full 3D structure of Nsp1 from I-TASSER (ID: QHD43415_1) [18,19]. The chain ID is missing in the I-TASSER 3D structures of Nsp1; therefore, PyMOL (Version 2.3.0 Schrödinger, LLC) Molecular Graphics System was used to add the chain ID for further analysis. I-TASSER is most commonly used server, accurately predicting and modeling the protein 3D structure from sequences. The server also provides supporting information, including ligand sites, model validations, and many important residual modeling graphs. The server has already modelled the full-length proteins of SARS-CoV-2 using NCBI reference data (NC_045512) (GenBank MN908947). I-TASSER contains a full-length SARS-CoV-2 protein that is freely available to the academic community (http://zhang.bioinformatics.ku.edu/I-TASSER, accessed on 30 December 2020). The full-length structure of Nsp1 (ITASSER ID: QHD43415_1) was uploaded to the DynaMut server along with the mutations at each amino acid location one by one. This was done in order to clearly identify the individual effects on the thermodynamic properties of Nsp1. 

### 2.5. Mutations’ Effects on Nsp1 Stability and Flexibility 

Mutations’ effects on Nsp1’s structural stability and flexibility were computed using the online DynaMut server [20]. Using the server, we implemented the normal-mode methods for the prediction of mutation effects that could be used to analyze the mutations’ effects on protein stability and flexibility following vibrational entropy changes. The impact of a mutation is predicted through normal-mode dynamics and graph-based signatures. This approach outperforms others (*p*-value < 0.001) and the results are also displayed in a good graphical resolution.

## 3. Results

Among the total whole-genome sequences, a large number of mutations were detected in Nsp1, present in the genomes of different geographic regions (Appendix A). All of these mutations were detected using the CoVsurver application (https://www.gisaid.org/epiflu-applications/covsurver-mutations-app/, accessed on 30 December 2020). The server is user-friendly and displays the detected mutations in the front column of each protein. The highest frequency of mutations was detected in USA genomic isolates (*n*- = 233), followed by England (*n* = 201), Australia (*n* = 50), and India (*n* = 27). However, the level of transmission and pathogenicity is largely unknown in relation to these mutations. A total of 933 non-synonymous and one synonymous mutation were detected in all 180-amino-acid residue positions of Nsp1 (Appendix A), whereas the frequency of each mutation was different at each of the individual positions. The highest substitution was observed at positions 84 (*n* = 10) and 85 (*n* = 10), followed by amino acid positions 17 (*n* = 9), 22 (*n* = 9), and 50 (*n* = 9). The most common mutation was R24C (*n* = 1122), followed by D75E (*n* = 890), D48G (*n* = 881), H110Y (*n* = 860), and D144A (*n* = 648) (Table 1). These mutations were detected in the flexible loop of the Nsp1 protein. The frequency of the other mutations was also significant (Appendix A). However, the frequency of the N-terminal domain was higher than that of the C-terminal domain of Nsp1.

### Effect of Mutations on Nsp1′s Stability and Flexibility

Mutations often affect the protein’s structural properties, including its stability and fluctuations. All of the detected mutations were submitted to the DynaMut server for stability and flexibility analysis. Among the 933 non-synonymous mutations, 529 exhibited a destabilizing effect. Similarly, a gain in flexibility was detected in 542 (Table 2, Appendix A). Amino acid types were considerably different in mutant (MT) Nsp1 when compared with the wild type (WT) (Figure 1). This difference might be useful in Nsp1′s function and stability, affecting SARS-CoV-2′s pathogenicity and virulency. The most common mutation (R24C), present at the N-terminal end, exhibited a destabilizing effect. Among the most common mutations, eleven exhibited increased flexibility when compared with WT Nsp1 (Table 1), in which the majority were present at the C-terminal domain. The frequencies of these mutations varied from 126 to 1122. Similarly, a cluster of mutations, present at position 56 to 75, demonstrated decreased fluctuation.

The globular domain in Nsp1 (Figure 2), in terms of gaining or losing flexibility in a particular residue, is shown in Figure 3. For the R24C mutant Nsp1, flexibility was altered in the region containing aa residues E65, L64, Q63, P62, C24, D16, and V14. Similarly, in MT D48G Nsp1, a gain in molecular flexibility was detected at aa residues S40, R43, Q44, K47, and G48. A decrease in molecular flexibility was detected in MTs D75E Nsp1 and H110Y (Figure 3), where the blue regions indicate a rigidification and red represents a gain in flexibility. These changes in fluctuation upon non-synonymous mutations might have effects on its normal function in terms of blocking host translation and evading the immune system. 

The collective differences in amino acid types in WT and MTs Nsp1, detected in all genomic sequences, are shown (Figure 1). Given the critical role that Nsp1 plays in the SARS-CoV-2 life cycle, amino acid differences in WT and MT may alter the virus’ virulency, pathogenicity, and infectivity. The majority of the most frequently occurring mutations were detected in the loop region (Figure 2). Among these, the highest difference was observed in glutamic acid, glycine, isoleucine, and valine. 

The most unstable region behind a mutation was detected at aa121 to 124 (Table 3). Numerous mutations at these positions, with various frequencies, were detected. Most of these MTs seem to exert a destabilizing effect, except V121I, V121W, and L122I, present at very low frequencies. 

## 4. Discussion

Nsp1 has the capacity to redirect proteins that are vital in host translational mechanisms. It is known as the leader protein, also referred to as the “host shutoff factor” and “cellular saboteur” [13,21,22,23,24,25,26]. Although its role is important in SARS-CoV-2’s pathogenesis and evolution, the effect of mutations on Nsp1 activity needs to be investigated for better management of the SARS-CoV-2 infection. Nsp1 likewise prevents the host from collecting an antiviral arsenal [27]. Mutations may affect the virus’ pathogenicity, infectivity, and transmission [28,29,30,31]. Studies have reported that Nsp1 interferes with the host immune response [8,9,10,11] and its deletion results in preventing cultured cells from becoming infected [6]. Moreover, mutations in Nsp1 prevent its release from the nascent polyprotein ORF1a, limiting the viability of the virus. The Nsps are important for vaccine development and as potential drug targets [32,33] because of their role in viral genome replication, disrupting the host cell environment, and interfering with the immune system and signaling [34,35]. SARS-CoV-2 is evolving rapidly and genetic diversity seems to accumulate in a short duration. As per estimations, the substitution mutation rate was high initially [36] and there is a possibility that mutations accumulated during the pandemic could produce measurable effects on the infected population or complicate epidemic control efforts, highlighting a need to monitor the genetic diversity and epidemiology of SARS-CoV-2 throughout the pandemic.

Among all the Nsp1 mutations, 232 and 201 different types were detected in genomic isolates from England and USA, respectively, 50 from Australia and 27 were detected in Indian isolates (Appendix A). According to global statistics, the total number of cases reported till 7 August 2021 in the USA, the UK, Australia, and India was 39.5 million (M), 6.83M, 0.56M, and 32.9M, respectively. However, the frequency of Nsp1 mutations from each of these countries was different due the different amounts of whole-genome-sequenced data available in GISAID. The highest frequency of mutations for R24C (*n* = 1122) and D48G (*n* = 881) was detected in SARS-CoV-2 genomic isolates from England; on the contrary, D75E (890) was detected in New Zealand isolates. 

Nsp1 obstructs the ribosomal entry site to avert host mRNA binding because it acts as a translation inhibitor via its C-terminal [21]. A SARS-CoV-2 genomic variation has been distinguished with a 9-base-pair deletion in position 686–694, relating to amino acids KSF at position 241–243. This adjusts Nsp1, with a resulting inference in SARS-CoV-2’s pathogenesis [37]. Recently, a genome from Bangladesh was found to have some novel mutations, including V121D, which destabilizes Nsp1, inactivating the antiviral system mediated by type 1 interferons. Therefore, this MT Nsp1 may be used in attenuated vaccines. Mutations often affect the thermodynamic properties of proteins [38,39,40,41]. In the current study, we also detected numerous non-synonymous mutations (V121F (*n* = 14), V121A (*n* = 1), V121D (*n* = 1), V121I (*n* = 13), V121P (*n* = 1), and V121W (*n* = 1)). All these mutations exhibited a destabilizing effect, except for V121I and V121W (Table 3). However, the immune response should be investigated before its subjection to vaccine development. We detected 933 non-synonymous mutations for the first time, among which the majority exhibited a destabilizing effect and increased flexibility. These stability effects behind mutations in Nsp1 might be useful if they weaken the virus’ replication competency. In other words, the MT Nsp1 may have a very weak immune response, resulting in a severe infection and pathogenicity. However, mutations’ effects must be experimentally verified before their application as attenuated vaccines. 

As reported in a recent study, the Nsp1 portion of SARS-CoV-1 and SARS-CoV-2 is well conserved (84% identity). This high conservation (aa10-126) makes up the crystal structure of Nsp1 (86%) [6]. However, in the current study, we observed that all of the crystal structure residues harbor numerous mutations in each position (Appendix A). Forty-two different non-synonymous mutations have been detected in the loop (aa9-16). It is well known from previous studies that the loop region of the protein plays a critical role in many interactions. Therefore, mutations in these locations may have a particular effect on their structure and thermodynamic properties [42,43,44,45,46]. 

Similarly, mutations in loop 2 and 3_10_ helixes (aa20-35) were also detected (Appendix A), exhibiting a destabilization effect in the majority of the mutation positions. The most commonly observed mutation was detected at position R24C, present in 3_10_ helices (aa20-35). It demonstrated a destabilizing effect and a gain in Nsp1 fluctuations. As previously reported, the η1 (3_10_ helix) is present only in the SARS-CoV-2 structure (aa23-25) and there are no apparent variants in the SARS-CoV-2 3_10_ helix that could stabilize them. However, here, we report that this is a hotspot region for the most frequently occurring mutations (R24C, R24H, R24S, and R24E). Many mutations (R24C, R24H, R24S, R24E, R24V, and R24A) are also present in the 3_10_ helix, among which R24C is present with the highest frequency (*n* = 1122) in all Nsp1 mutations, with a destabilizing effect. However, R24S and R24A MTs Nsp1 exhibited a stabilizing effect and a gain in flexibility. All of the 3_10_ helix mutations demonstrated an increase in flexibility, except for R24H MT. The effects on flexibility and stability need to be investigated through further experimental approaches for better management of drug design against Nsp1. Point mutations may result in structural perturbations, which may have a pronounced effect on protein flexibility over long distances [47,48].

Recently, four novel mutations have been detected in the genome of SARS-CoV2, among which V121D substitution exhibited a destabilizing effect on Nsp1 [49]. Similar to this study, we detected 18 non-synonymous mutations at positions aa 121 to 124, where all variants exhibited a destabilizing effect, except for V121I, V121W, and L122I. A suitable MT Nsp1 of SARS-CoV-2, which has a major effect on the virus’ replication competency, may be promising for use as an attenuated vaccine. The destabilizing effect of MT Nsp1 may be investigated to determine whether the mutations weaken the virus’ replication competency or the immune response. In the former case, the MTs may be used as vaccines, while in the latter case, the virus’ pathogenicity might be increased. A number of mutations present in the loop and 3_10_ helices exhibited a destabilizing effect in Nsp1 (Appendix A) and this may be investigated for application against SARS-CoV-2 infection. Geography-specific vaccines and drugs may be designed after the careful analysis of the variants in the target proteins for the better management of SARS-CoV-2 infections. According to a recent study, deletion variants in SARS-CoV-2 clinical isolates with related deletions in Nsp1 also induce lower IFN-b responses in infected cells. 

Although we screened a large number of genomes for mutations in Nsp1, there are some limitations in the current study. Mutations have not been screened in other Nsps, which might be useful to analyze their association with the virulency of SARS-CoV-2. Secondly, the effects of mutations in functionally important sites of Nsps need to be investigated through experimental approaches.

## 5. Conclusions

Mutations are gaining importance due to their role in potential viral pathogenicity and alteration in the host immune response. In the current study, mutations in nearly all amino acid positions of Nsp1 showed that the virus passes through critical stages. This protein exhibited various degrees of stability and flexibility behind mutations in the important loop regions, which might have some consequences for the virus’ replication competency and the host immune response. Mutations at amino acid positions 121 to 124 exhibited a high frequency of destabilization, with the exception of V121I, V121W, and L122I, present in very low frequencies. 

Further experimental investigations are needed to explore the effect of Nsp1 mutations on virus replication and the immune response that might be useful in diagnosis, and also in vaccine development. More longitudinal studies will further unveil the relationships between Nsp1 mutations and virus replication competency. The host immune response may also be analyzed in infected patients with these MTs. The current study provides useful information for accurate diagnosis and drug design.

## Figures and Tables

**Figure 1 pathogens-10-01285-f001:**
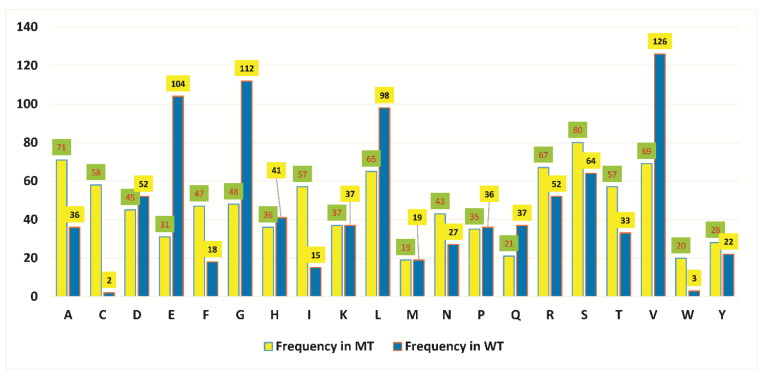
Frequency of amino acids in wild-type (WT) and mutant (MT) Nsp1. Amino acids E, G, L, and V are significantly different in WT and MT. Blue represents WT and yellow represents MT.

**Figure 2 pathogens-10-01285-f002:**
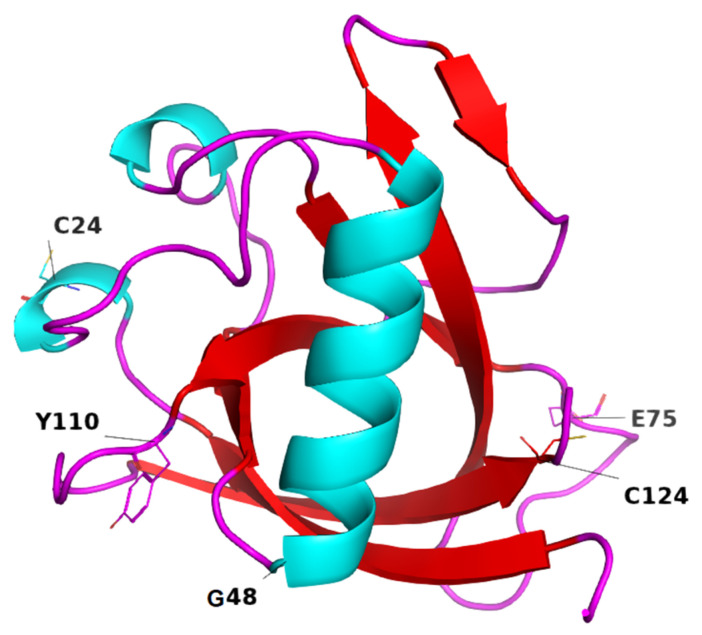
Most frequent mutations in Nsp1 of SARS-CoV-2. Frequency of R24C (*n* = 1122), G48 (*n* = 881), D48G (*n* = 881), H110Y (*n* = 860), and R124C (*n* = 126) is shown, present in the loop regions.

**Figure 3 pathogens-10-01285-f003:**
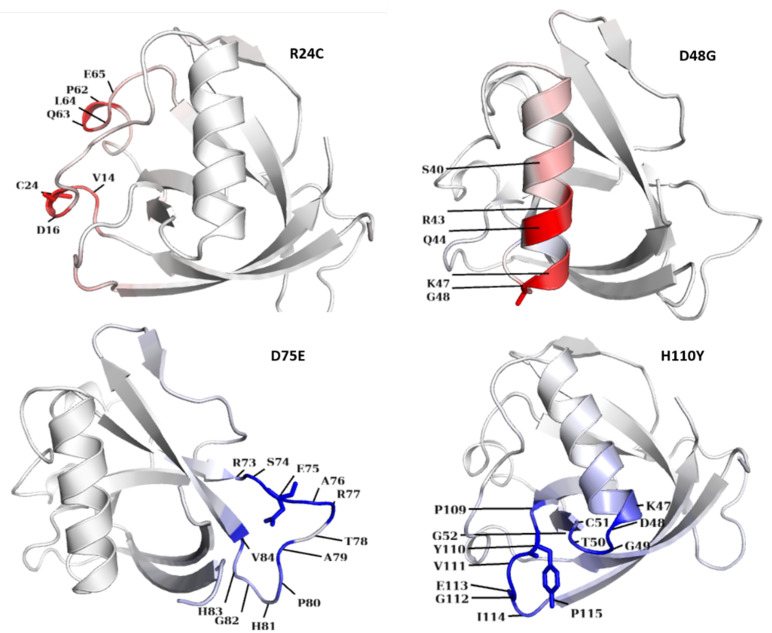
Most common mutations in the Nsp1 globular domain and their effect on flexibility. Residues at red region of Nsp1 show gain in flexibility in common mutations. Blue shows gain in rigidity. Residues at both types of regions are shown. Mutations’ sites and their surrounding residues are labelled, showing their flexibility level.

**Table 1 pathogens-10-01285-t001:** Most common mutations in the Nsp1 of SARS-CoV-2.

Mutation	Frequency	* ΔΔG: kcal/mol	Stability	* ΔΔSVib ENCoM kcal.mol^−1^.K^−1^	Flexibility
R24C	1122	−0.218	Destabilizing	0.158	Increase
E37D	237	−0.844	Destabilizing	0.348	Increase
E37K	309	−0.243	Destabilizing	0.360	Increase
H45Y	143	1.262	Stabilizing	−0.217	Decrease
D48G	881	−0.217	Destabilizing	0.024	Increase
V56I	113	0.954	Stabilizing	−0.228	Decrease
E57G	212	0.332	Stabilizing	−0.112	Decrease
V60I	188	0.617	Stabilizing	−0.192	Decrease
D75E	890	0.31	Stabilizing	−0.035	Decrease
D75G	115	0.676	Stabilizing	−0.066	Decrease
M85V	184	−0.105	Destabilizing	0.687	Increase
E87D	407	−1.088	Destabilizing	0.395	Increase
L92F	199	0.143	Stabilizing	−0.002	Decrease
H110Y	860	0.452	Stabilizing	−0.060	Decrease
K120N	418	−0.320	Destabilizing	0.129	Increase
R124C	126	−1.116	Destabilizing	1.056	Increase
N126S	209	−0.692	Destabilizing	0.177	Increase
D144A	648	−1.411	Destabilizing	0.415	Increase
S166G	158	1.416	Stabilizing	0.225	Increase

* ΔΔG: Total energy, ΔΔSVib: Vibrational entropy energy.

**Table 2 pathogens-10-01285-t002:** Frequency of the stability and flexibility effects, predicted for all Nsp1 MTs.

Stability	Frequency
Destabilizing	529
Stabilizing	404
Total	933
**Flexibility**	
Decrease	391
Increase	542
Total	933

**Table 3 pathogens-10-01285-t003:** Mutations at positions 121-124 and their effect on Nsp1’s stability.

S. NO	Mutation	Frequency	* ΔΔG: kcal/mol	Stability
1.	V121F	14	−0.463	Destabilizing
2.	V121A	1	−2.194	Destabilizing
3.	V121D	1	−1.469	Destabilizing
4.	V121I	13	0.375	Stabilizing
5.	V121P	1	−1.847	Destabilizing
6.	V121W	1	1.136	Stabilizing
7.	L122I	3	0.184	Stabilizing
8.	L122V	2	−2.541	Destabilizing
9.	L122F	21	−0.577	Destabilizing
10.	L122P	1	−1.106	Destabilizing
11.	L122R	1	−0.788	Destabilizing
12.	L123F	14	−0.784	Destabilizing
13.	L123W	1	−1.189	Destabilizing
14.	L123E	1	−2.835	Destabilizing
15.	L123I	1	−0.175	Destabilizing
16.	R124C	126	−1.116	Destabilizing
17.	R124H	3	−0.34	Destabilizing
18.	R124V	2	−0.096	Destabilizing

* Free energy difference.

## Data Availability

The datasets in the current study are available as Appendix A and can be freely accessed from GISAID (https://www.gisaid.org/).

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
