# Peer review of "Emerging Mutations in Nsp1 of SARS-CoV-2 and Their Effect on the Structural Stability"

_pathogens, 2021, doi:10.3390/pathogens10101285_

Round 1
Reviewer 1 Report
My Decision is Minor revision
These authors have analyzed Nsp1 mutations in covid-19 patients. NSP1 inhibits the host gene expression and also evades the immune system. It has therefore been proposed as a target for vaccine development and as a drug target. These findings show that Nsp1 is a mutation point that might be useful to exploit in order to decrease CoV-2’s pathogenecity.
The current study aimed to find mutations in the Nsp1 of CoV-2 and their effect on protein stability and flexibility. These authiors screened 295000 complete genomes and compared them with the Wuhan-Hu-1 reference genome (Accession NC_045512) using CoVsurver app.
The sequences harbor 934 mutations in the entire coding region of Nsp1. The most frecuent mutations in the 180 amino acid Nsp1s were R24C (n=1122), D75E (n=890), D48G (881), H110Y (n=860) and D144A (n=648).
Among the 933 mutations, 529 exhibited a destabilizing effect.
Please, explain the exactly reason of this destabilizing effect and the possible consequence in terms of infection and disease.
Please, explain why the most frequent mutations are detected in the loop regions.
-Which are the possible link between these identified mutations and risk of covid-19 infectivity and pathogenesis
-how drug targets could improve the infection and improve the diagnostic accuracy in patients?
-Shall you argue about mechanisms by which these mutations affects Nsp1s stability and possibvle altered host innate immune by COV-2 infection in patients?.
-How these mutations could affect clinical eficacy of different kind of vaccines against Covid-19-infected patients?
Sugestions to the author
Introduction ¨Nsp1 is the N-terminal product, cleaved by papain-like proteinase (PLpro) from the polyprotein, promoting the host mRNA degradation, blocking translation and interfering with the innate immune response [6–9,9]¨. Please, give more details about how Nsp1 cleaved could affect innate immune responses in patients; add it to the introduction since this point is important in the context of Covid-19 infection.
Method section…¨The CoV-2 genome sequences were retrieved from the global science initiative shar-ing all influenza data (GISAID) (Dec, 2019 to Dec, 2020) (https://www.gisaid.org/) [10) and Wuhan-Hu-1 (Accession NC_045512) using the CoVsurver application (https://www.gisaid.org/epiflu-applica-tions/covsurver-mutations-app/). The I-TASSER server has already modelled the full-length proteins using the NCBI reference data (NC_045512) (GenBank MN908947). I-TASSER developed a full-length COV-2 protein that is freely available to the academic community (http://zhang.bioinformatics.ku.edu/I-TASSER)¨
Please, give more extensive information about how to use these programs and websides for a general audience without knolegment in these genetic database and include this information within the methods section. Thanks¡
Results
All of the mutations were analyzed using the CoVsurver application (https://www.gisaid.org/epiflu-applications/covsurver-mutations-app/).
The same within result section. Please, explain how you analyze data in webside and include it within this section
-Please, fig 3 is difficult to see. Please, improve it (nmbers are difficult to see in this figure).
Discussion
Please, explain possible differences in terms of infection by Covid-19 between patients from England, USA, Australia and Indian (Supplementary file S1). They indicates…¨ Among all the Nsp1 mutations, 232 and 200 of different types were detected in ge-nomic isolates Among all the Nsp1 mutations, 232 and 200 of different types were detected in ge-nomic isolates from England and USA, 50 from Australia and 27 were detected in Indian isolates (Supplementary file S1). (Supplementary file S1)¨.
Please, try to explain why these differential mutations ( Destabilizing or stabilizing) could affect Covid-19 infection. Precise if these mutations can differentially affect the inhate and adapative immune responses in Covid-19-infeceted patients. Include all these comments within discussion section in terms of immune system alterations and vaccine responses in patients. How these Destabilizing or stabilizing mutations could affect Covid-19 the progression of diseases in patients with pneumonia?
Thanks a lot.
Author Response
Dear Sir, We appreciate your comments on our manuscript. We tried our level best in adressing all of your comments. Hope these will be acceptable.

Reviewer 2 Report
In this study, Mou et al. screened 295.000 complete SARS-CoV-2 genomes (having as reference genome the Wuhan-Hu-1 genome) and identified 934 mutations in the entire coding region of Nsp1. The authors did an in silico analysis concerning the destabilizing effect and the loss or gain in flexibility that is calculated to occur because of these mutations. However, those results remain to be confirmed through experimental approaches. Finally, the authors present the most frequently occurring mutations and highlight the importance of the study as Nsp1 seems to be a mutation point that could be exploited in order to decrease CoV-2’s pathogenecity, or to serve as scaffold to design geographically strain-specific drugs and/or vaccine in order to build a new strategy against CoV-2 infection.
Although, the study is well presented and could be very interested for the readers, the authors should consider to add and clarify in the text how the specific findings will lead to the design of new geographically strain-specific drugs and/or vaccine. More specifically, they could clarify what would mean for the SARS-CoV-2 and its inhibition mechanism of the host gene expression, if the flexibility in the loop of NSP1 is increased or if there is a destabilizing effect and how these effects could be used for further investigating the designing of a drug or a vaccine that could target the NSP1. This way, it will be easier for the readers to follow the hypothesis of the research.
Moreover, are there any data on the specific SARS-CoV2 genomes on the virulence or efficacy of the specific virus that could be correlated with their mutation? If so, please add these data to the manuscript.
Please consider the following minor changes:
Abstract:
1) Are there 934 or 933 mutations. Pleace change to be in accordance in all the article: "The sequences harbor 934 mutations in the entire coding region of Nsp1. ...... Among the 933 mutations, ...."
2) Add (n=...) in all parentheses: "The most frequently occurring mutations in the 180 amino acid Nsp1s were R24C (n=1122), followed by D75E (n=890), D48G (881), H110Y (n=860) and D144A (n=648)."
3) Replace "show" with "imply": "These findings imply that Nsp1 is a mutation point that might be useful to exploit in order to decrease CoV-2’s pathogenecity."
Background:
4) Add then to the sentence: " It then degrades the host mRNAs by prompting an endonucleolytic cleavage near the 5´-untranslated region [2]. "
5) Change the following sentence in order to have a verb also in the second half concerning β2, βε, β4, β5, β6 and β7: ". The β1 consists of eight residues (aa13-20), β2 (aa51-54), β3 (aa68-73), β4 (aa84-92), β5 (aa95-97), β6 (aa103-109), and β7 (aa117-123)."
6) Change the sentence as follows: "In a recent study, the structure of SARS-CoV-2 Nsp1(13-127) was shown to have a feature arrangement with six-stranded (n = 6) β-barrel antiparallel except strands β1 (Q15 – V20) and β2(C51–V54) [5]."
7) Delete the one 9 for the references: "...mRNA degradation, blocking translation and interfering with the innate immune response [6–9,9]"
8) Add a reference at the end of the sentence: "Furthermore, mutations in the coding region of Nsp1 prevent its release from the nascent polyprotein ORF1a, limiting the viability of the virus."
Results:
9) Add (n=...) in all parentheses: "The most common mutations were R24C (n=1122), followed by D75E (n=890), D48G (881), H110Y (n=860) and D144A (n=648)"
10) "The frequency in the mutations of the N-terminal ......"
11) delete one gap between the sentences: "....... ....S40, R43, Q44, K47 and G48. A decrease in molecular flexibility has been observed in MTs D75E and H110Y (Figure 3)..."
12) "These MTs might be used for vaccine development after experimentsl verification."
Discussion:
13) Please correct the following sentence in order to be more comprehensible: "The Nsp’s are important for the viral genome replication and for vaccine development as potential drug targets [24,25]."
14) Add (n=...) in all parentheses: ". However, the highest frequency of mutations (R24C (n=1122), D48G (881)) was detected in SARS-CoV-2 genomic isolates from England except D75E (890) which was detected in New Zealand isolates."
15) Delete one gap between the sentences: "This adjusts Nsp1 with a resulting inference in SARS-CoV-2 pathogenesis [26]. Recently, a genome from Bangladesh was found to have some novel mutations including V121D which..."
16) Add "(Supplementary file S1)" instead od "(S1)" throughout the article: "Similarly, mutations in loop 2 and 310 helixes (aa20-35) were also detected (S1) were also present in sufficient number, exhibiting a destabilizing effect in the majority of variant positions as shown in supplementary file (S1)."
17) "310 helix" instead "310 helix" throughtout the article: "As previously reported, the η1 (310 helix) ...."
18) Delete however from sentence as it starts with although: "Although we screened large number of genomes for mutation in Nsp1, there are some limitations of the current study."
Author Response
Dear Sir,
We appreciate your comments on our manuscript. We tried our level in addressing of all your comments. Hope these will be acceptable.

Reviewer 3 Report
In this manuscript “Emerging mutations in nsp1 of SARS-CoV-2 and their effect on the structural stability”, Mou et al. analyze the sequence of severe acute respiratory syndrome coronavirus 2 (SARS-CoV-2) non-structural protein 1 (nsp1) from 29,500 genomes available from the GISAID database. Authors identified 934 mutations in SARS-CoV-2 nsp1 being R24C, D75E, D48G, H110Y, and D144A the most frequent mutations. Authors also conducted an in silico assessment on how these mutations could affect nsp1 stability and flexibility. Authors speculate that the identified mutations affect nsp1’s ability to inhibit host gene expression and, therefore, could have an effect on SARS-CoV-2 pathogenicity. However, there are no experimental evidence that these mutations affect the ability of nsp1 to inhibit host gene expression and/or innate immune response, or its stability, to support this conclusion.
Major comments:
- The role of the identified mutations in nsp1’s function are based on in silico prediction. Authors could easily assess if the identified mutations affect the ability of nsp1 to inhibit host gene expression and/or innate immune responses using previously described cell-based assays.
- Likewise, authors could easily assess if the identified mutations affect nsp1 stability by simply assessing protein expression in cells transfected with plasmids expressing wild-type and mutant nsp1 proteins, and Western blot analysis.
- The authors hypothesize that the identified mutations most likely have an effect on viral fitness and pathogenesis. However, without data with recombinant viruses containing the identified mutations and a direct comparison with wild-type SARS-CoV-2, it would be difficult to demonstrate this hypothesis.
- Several sentences indicates that nsp1 evade host innate immune responses. This is most likely a consequence of the ability of nsp1 to inhibit host gene expression rather than an specific function of nsp1.
- There is a repetitive mentioning of targeting nsp1 for vaccine development or antiviral targeting. However, there is no specific information on how results from this study could help vaccine or antiviral development.
- Figure 3 legend is too small to see the amino acid positions in the nsp1 structure.
Minor comments:
- The authors should revise the document and keep consistency with the nomenclature of some of the abbreviations (e.g. SARS-CoV-2 vs COV-2).
- Likewise, the authors should revise the manuscript to correct some typos/spelling mistakes.
- Authors should also carefully revise the references for accuracy and proper citations. Moreover, authors should include citations to support some of their statements.
Author Response

(The authors gave the same response as above.)

Reviewer 4 Report
In Pathogens manuscript ID # pathogens-1336058, Mou et al. present a computational analysis of observed mutations in SARS-CoV-2 nonstructural protein 1 (Nsp-1) sourced from global public databases containing complete viral genome sequences. They identify mutations within Nsp-1 and exam their theoretical effect on protein structure using a computer modeling program. While the predictions of the impact of the identified mutations on protein structure are a strength of the current study, the lack of supporting biological data limit the conclusions and interpretation of those identified substitutions.
The major flaw with the current manuscript is that even the most frequent mutation occurs at a very low frequency, and the authors rely strictly on computer software modeling to investigate the impact of these mutations on Nsp-1 function. As described below, because the authors rely strictly on computer modeling, it is possible that none of these very infrequent mutations result in a replication competent virus. This seems to be supported by their conclusion that most mutations identified are predicted to result in destabilization of the Nsp-1 protein. Therefore, in the current state, the findings lack biological significance.
Minor Concerns:
- A defined and consistent abbreviation for SARS-CoV-2 should be used.
- The manuscript requires extensive proofreading and editing for language, syntax and scientific language errors that detract from the overall focus.
Major Concerns:
- The authors provide no evidence that the mutations identified in Nsp-1 result in replication competent virus. Thus, it is entirely possible that all observed mutations result in dead-end virus and are therefore of no significance. In fact, even the most frequent substitution (R24C), was present in just 1122 of the 295,000 genomes analyzed (0.04% frequency).
- Perhaps some of these mutations have been identified in a specific clinical isolate(s) that has been characterized or examined in vitro, demonstrating that the virus is replication competent? This could provide support for the authors that the mutations are significant.
- Or perhaps the authors could map the mutations by geographic region? If the mutations cluster in an epidemiologic map, then that might provide support that the mutations are significant in terms of viral spread?
Author Response

(The authors gave the same response as above.)

Round 2
Reviewer 3 Report
The authors have tried to address the major criticisms from the first submission of the document. However, and sorry to say, they have not provided with scientific evidences that the identified nsp1 mutations have an effect on inhibition of host gene expression and/or protein stability as suggested in my previous review to support their conclusions. Thus, the document is still speculative and lack scientific proof. The authors should at least make an effort to test experimentally that the identified mutations affect the ability of nsp1 to inhibit host gene expression and protein stability to support their conclusions.
Author Response
Author reply: We appreciate the reviewer comments. The scientific evidences that the identified nsp1 mutations have an effect on inhibition of host gene expression was not found in the literatures and also the experimental evidence is not available in the current situations. The experimental evidence is not possible to be provided in the current study because it needs a long time and cost to performed in the well-furnished p3 laboratory. However, based on our results in this project one can easily design the experimental knockout research project to investigate the Nsp1 mutations effect on host gene expression. The conclusion in our results has been supported with bioinformatics approaches (DynaMut) and experiments may be performed in future. We predicted the effect of Nsp1 mutations on protein structure using the most widely used DynaMut. The effect on host gene expression through experimental approaches, using our current results. We hope that this reply will be acceptable.
Research design, methods, results and conclusion has been improved.
English errors were checked again.
Reviewer 4 Report
The inclusion of the expanded details of the Cell paper, finding that a deletion in Nsp1 results in reduced cytokine responses suggests that the identified potential mutants could result in immune escape. In addition, it provides some evidence that these low-frequency mutations have biological significance and increases the significance of the results. However, it is important to note that the database used to identify the described mutants was generated from PCR amplification of clinical samples. This does not demonstrate that the detected sequences represent replication competent viruses, or impact immune responses as described for the other Nsp1 mutations in the cited Cell paper. This can only be accomplished through reverse genetic approaches and subsequent virus replication studies. In addition, there is no evidence that the mutants are being selected for in geographic locations. But overall, the revised draft is significantly improved.
Minor Problems:
Line 258 is an incomplete sentence.
Author Response
Comments and Suggestions for Authors
The inclusion of the expanded details of the Cell paper, finding that a deletion in Nsp1 results in reduced cytokine responses suggests that the identified potential mutants could result in immune escape. In addition, it provides some evidence that these low-frequency mutations have biological significance and increases the significance of the results. However, it is important to note that the database used to identify the described mutants was generated from PCR amplification of clinical samples. This does not demonstrate that the detected sequences represent replication competent viruses, or impact immune responses as described for the other Nsp1 mutations in the cited Cell paper. This can only be accomplished through reverse genetic approaches and subsequent virus replication studies. In addition, there is no evidence that the mutants are being selected for in geographic locations. But overall, the revised draft is significantly improved.
Author response: We appreciate the reviewer valuable comments. English language and changes has been revised
Minor Problems:
Line 258 is an incomplete sentence.
Author response: We appreciate the reviewer valuable comments. The sentence has been completed.

Round 3
Reviewer 3 Report
The authors still do not want to test in cell-based systems, that only require the use of BSL2 facilities, if the identified nsp1 mutations have an effect on inhibition of host gene expression and/or protein stability. I still think these experiments will increase the significance of the manuscript and will allow to prove that the mutations affect nsp1 function and expression, respectively.